# A single donor is sufficient to produce a highly functional in vitro antibody library

M. Frank Erasmus [1,4], Sara D'Angelo [1,4], Fortunato Ferrara [1,4], Leslie Naranjo[1], André A. Teixeira[2], Rebecca Buonpane[3], Shaun M. Stewart[3], Horacio G. Nastri [3] & Andrew R. M. Bradbury [1✉]

Antibody complementarity determining region diversity has been considered to be the most important metric for the production of a functional antibody library. Generally, the greater the antibody library diversity, the greater the probability of selecting a diverse array of high affinity leads. According to this paradigm, the primary means of elevating library diversity has been by increasing the number of donors. In the present study we explored the possibility of creating an in vitro antibody library from a single healthy individual, showing that the number of lymphocytes, rather than the number of donors, is the key criterion in the production of a diverse and functional antibody library. We describe the construction of a high-quality phage display library comprising $5 \times 10^9$ human antibodies by applying an efficient B cell extraction protocol from a single donor and a targeted V-gene amplification strategy favoring specific antibody families for their improved developability profiles. Each step of the library generation process was followed and validated by next generation sequencing to monitor the library quality and diversity. The functionality of the library was tested using several therapeutically relevant targets for which a vast number of different antibodies with desired biophysical properties were obtained.

[1] Specifica Inc, Santa Fe, NM, USA. [2] New Mexico Consortium, Los Alamos, NM, USA. [3] Incyte Research Institute, Wilmington, DE, USA. [4] These authors equally contributed: M. Frank Erasmus, Sara D'Angelo, Fortunato Ferrara. ✉email: abradbury@specifica.bio

Monoclonal antibodies are now the most successful class of therapeutics, representing seven of the top ten best-selling pharmaceuticals. Effective development of therapeutic antibodies demands high-quality molecules to meet strict manufacturing requirements and success in clinical applications. The first step in antibody discovery is the identification of potential candidate binding targets of interest, with two main methods used: hybridoma generation (or more recently single-cell cloning methods) from the immunization of normal or transgenic animals, and display methods based on naive or immune libraries. The source of diversity for libraries from which antibodies are selected has taken many formats: immune repertoires, generated from immunized animals or humans seropositive for a target; natural naive, derived from the B lymphocytes of non-immunized donors[1–3]; synthetic, in which diversity is derived from oligonucleotides[4–9]; semi-synthetic, combining synthetic and natural diversity[10]; and PCR-based recombinatorial approaches in which natural CDR diversity is shuffled within a fixed scaffold[11,12].

Specific binders are selected from libraries displayed using platforms such as phage/phagemid[1,13,14], yeast[15,16], ribosome[17,18], and mammalian display[19–21], with phage and yeast display being the most commonly used. These methods couple phenotype to genotype: the gene encoding the displayed antibody is coupled to the antibody itself, allowing the cloning of antibody genes on the basis of the properties (e.g., binding activity) of the encoded antibodies. Once selected, the affinity or specificity of an antibody can be further improved by applying the same display methods to derivative libraries.

The key feature of any compound library, including antibody libraries, is diversity. In the case of antibody libraries, this is often measured by counting the number of clones obtained after library transformation. However, this is an inaccurate measure at best, plagued by uncertainties in colony counting and the underlying genetic diversity, which can often be significantly less than the number of transformants[3,22,23]. Diversity correlates with library functionality: the probability of discovering an antibody in the library with desired activity (e.g., affinity, functional activity, or therapeutic efficacy). In the case of antibody libraries, other desirable key features are "developability properties": stability and resistance to aggregation in solution, thermodynamic stability, and low self-interaction, among others. Therefore, scientists strive to make antibody libraries as large as possible by performing numerous transformations or, less commonly, applying site-specific recombination approaches[3,4,24]. However, unless the underlying genetic diversity is sufficiently large, generating more transformants will not necessarily increase library functionality. This is less of a problem for synthetic or semi-synthetic libraries, in which the theoretical diversity far exceeds the number of transformants that can be routinely obtained. For example, the theoretical diversity of the HuCAL Gold library[8] was estimated to be $10^{60}$, but the number of transformants, $10^{10}$, was the limiting factor. Although the assessment of genetic diversity is clearly an improvement over counting colonies, functional diversity is a more valuable measure, particularly in the case of synthetic libraries in which degenerate oligonucleotides may encode poorly folding, or polyreactive, antibodies, resulting in a larger chemical space that does not necessarily directly correlates with protein functionality.

For natural naive libraries, the general paradigm has been to increase the natural source of B-cell diversity by increasing the number of donors[23,25,26]. The question remains as to whether increasing the number of donors actually leads to improved antibody diversity and library functionality.

In recent papers examining library diversity with next-generation sequencing (NGS), estimated genetic diversity was far lower than anticipated[22,23,26,27]. This may be a consequence of cloning or PCR inefficiencies, inherently restricted underlying diversity (e.g., insufficient numbers of B cells or libraries skewed by excesses of memory or plasma cells), or problems with harvesting the natural diversity, by failing to collect and reproduce the entirety of the diversity present in the starting material. It is also possible that the focus on increased donor numbers is misplaced. In a recent paper from DeWitt and colleagues[28], using a combined flow cytometry/NGS analysis of B cells from healthy individuals, the authors inferred a $\sim 10^9$ diversity of circulating naive B cells in a donor at any given time, and proposed that naive B lymphocytes do not undergo (or undergo minimal) further cell division after VDJ recombination prior to encountering antigen. In fact, a library created from one of the greatest number of donors recorded to date (654)[23], with a large number of transformants ($3.1 \times 10^{10}$) had one of the lowest VH diversities when assessed by NGS ($10^5$), indicating the importance of other factors in library generation beyond donor number. Here we tested the hypothesis that the final number of lymphocytes used to generate a library, and not the number of donors, determines library diversity, and hence library functionality. The main assumption is that harvesting a large number of B cells from a single individual may be as effective as increasing the number of donors. Because the capture and cloning of a diversity of $10^9$ are well within antibody display state-of-the-art capabilities, we set out to determine whether, through the use of good library construction practices, we could produce a highly functional library from a single donor (single donor library or SDL), where the library is intended to maximize the diversity of V genes with favorable biophysical properties to isolate antibodies with therapeutic properties. For this reason, we used a restricted set of primers to bias amplification toward specific antibody families identified by Tiller and colleagues[29] for their improved developability profiles including aggregation, solubility, and protease degradation. Further, several other families show increased representation in the naive repertoire which we tried to capture in this library.

We hypothesized that through the combined use of leukapheresis technologies, lymphocyte paramagnetic bead purification, careful gene-family focused PCR amplification, and NGS feedback, we could create a highly diverse and functional antibody library from a single donor.

## Results

**Construction of a naive single donor library (SDL).** Here we describe the construction of a phage display library obtained from the population of CD19+ cells purified from a single individual (Fig. 1a). Mononuclear cells were isolated from a single full Leukopak (LP) derived from a healthy male donor via leukapheresis. An LP contains all the circulating lymphocytes from a single donor, with approximately 20 times more B cells than a single unit of blood (525 ml). We isolated $7 \times 10^8$ B cells from the LP by magnetic-activated cell sorting using CD19 antibodies[30]. Between 90% and 95% of purified cells displayed CD20 and CD37, surface markers for pan B cells and mature B cells, respectively (Fig. 1b). The antibody repertoire was captured by extracting the total B-cell RNA, purifying the mRNA, and generating cDNA by RT-PCR using equimolar amounts of primers for the human CHμ, Cκ, and Cλ regions (Fig. 1c)[31]. The use of CHμ primers was intended to bias the cDNA synthesis towards naive IgM B cells, rather than IgG memory cells, to enhance the sequence diversity and presence of naive antibodies[1]. The cDNA served as a template for the amplification of the variable light domains (IGKV and IGLV) and variable heavy domains (IGHV).

Based on sequence similarity human IGHV, IGKV and IGLV genes have been organized into 7, 6, and 10 families, comprising

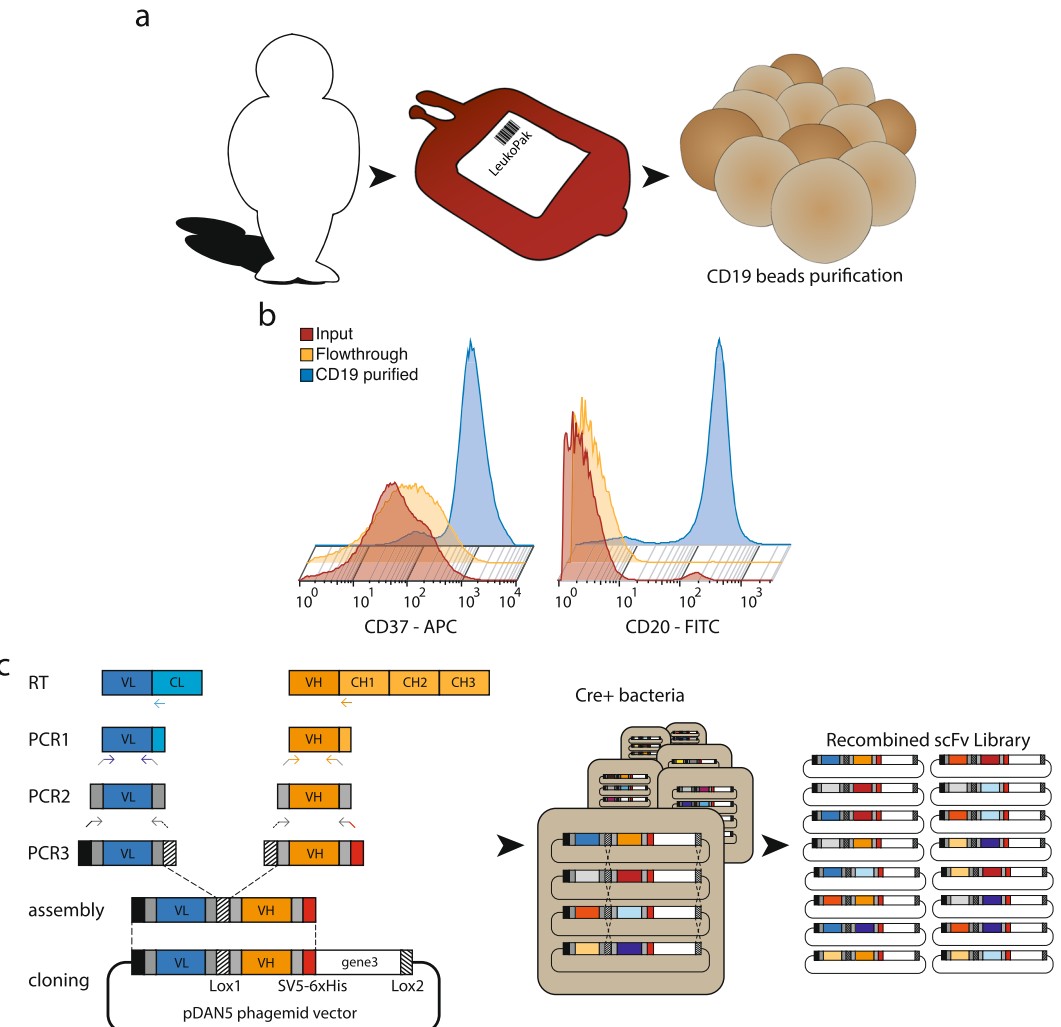

**Fig. 1 Process flow for B-cell purification, V-region amplification, and cloning into pDAN5 phage display vector. a** Schematics of B-cell purification of a single donor repertoire from LeukoPak. **b** FACS characterization of the CD19+ paramagnetically isolated cells for CD37+ (left) and CD20+ (right) staining. **c** Rescue of V domain diversity from donor and creation of the scFv phage antibody library: specific constant domain primers are used for the RT-reaction, followed by V domain amplification with IGKV/IGLV and IGHV-specific forward and reverse primers (PCR1). A second PCR introduces vector and linker overlaps (PCR2) that are exploited for the assembly of the scFv genes (PCR3). The genes are cloned into the pDAN5 phagemid vector by restriction enzyme digestion and transformed into bacteria. Upon superinfection with M13K07 helper phage, the phage is produced (primary library) and used to infect the recombinase positive bacteria (Cre+) at a multiplicity of infection of 1:200. The presence of incompatible lox sites between IGKV/IGLV and IGHV domains and downstream g3p (in black and white) allows recombination to occur between different VH and VL genes carried by different phagemid vectors. The process yields a phage population (secondary library) that has greater recombinatorial diversity than the primary library. A final phage production step, at a multiplicity of infection ≤1 restores the essential phenotype/genotype coupling (tertiary library/final library).

~50 IGHV[32–34], ~40 IGKV[35], and ~30 IGLV functional genes[36,37], respectively. To maximize library functionality, we used derivatives of a subset of commonly used primers[31] to avoid the amplification of germline scaffolds with undesired biophysical properties[29] (Tables 1 and 2). Each IGHV forward primer was used in a PCR with four different IGHV-J reverse primers. For the light chain, three IGLV-J (IGLV-J1, IGLV-J2/3, and IGLV-J7) and five IGKV-J forward primers were used in combination with the IGLV and IGKV reverse primers, respectively. For the IGHV4 family, a different amplification strategy was used, the goal of which was to rescue genes IGHV4-4 and IGHV4–59, and exclude IGHV4–31 and IGHV4–39, which are known to poorly perform during phage display selections[8,38]. Briefly, two overlapping primers were designed to be specific for the CDR1 region, directed outward toward the 5′ and 3′ ends of the domain. The obtained fragments were assembled to reconstitute the final restricted diversity IGHV4 domains.

More than 450 PCR reactions (PCR1 in Fig. 1c) were performed to amplify the IGHV and IGLV and IGKV domains (using the primers described in Table 1 and Table 2). For the IGHV domains, each of the 12 forward primers was individually used in combination with each of the 4 IGHV-J reverse primers in 48 PCR reactions. The resulting amplicons were mixed to reflect the IGHV-J distribution in natural repertoires (IGHV -J1 and 2: 5%; IGHV-J3: 10%; IGHV-J4, and 5: 60%; IGHV-J6: 25%)[39,40]. Similarly, the IGKV amplicons were obtained by individual PCR reactions using all combinations of 6 IGKV forward and 5 IGKV-J reverse primers and then mixed to reflect the IGKV-J distribution in natural repertoires (IGKV-J1: 29%; IGKV-J2: 26%; IGKV-J3: 10%; IGKV-J4: 26%; IGKV-J5: 9%)[40–42]. Likewise, the IGLV amplicons were generated by PCR using all the combinations of the 4 IGLV forward primers and 3 IGLV-J reverse primers and then mixed to represent the IGLV-J distribution in natural repertoires (IGLV-J1: 21%; IGLV-J2 + 3: 77%; IGLV-J7: 2%)[40].

**Table 1 IGHV domain developability and primer sequences.**

| ID | Desirable | Common in natural | Neutral | Poorly developable | Primer sequence |
|---|---|---|---|---|---|
| IGHV1a | 1–46, 1–69, 1–18 | | 1–08 | 1–02 | CAGGTKCAGCTGGTGCAG |
| IGHV1b | | 1–03 | | | CAGGTCCAGCTTGTGCAG |
| IGHV1c | | | 1–24, 1–69-2 | | SAGGTCCAGCTGGTACAG |
| IGHV1d | | | 1–45, 1–58 | | CARATGCAGCTGGTGCAG |
| IGHV2a | | | 2–05 | | CAGATCACCTTGAAGGAG |
| IGHV2b | | | 2–26, 2–70 | | CAGGTCACCTTGARGGAG |
| IGHV3a | 3–07, 3–15, 3–21, 3–48, 3–53, 3–74 | 3–09, 3–64 | 3–13, 3–20, 3–43, 3–49, 3–66, 3–72, 3–38-3 | 3–73 | GARGTGCAGCTGGTGGAG |
| IGHV3b | 3–11, 3–30, 3–33 | | 3–30-3, 3–30-5 | | CAGGTGCAGCTGGTGGAG |
| IGHV3c | 3–23 | | | | GAGGTGCAGCTGTTGGAG |
| IGHV4 | 4–04 | 4–59, 4–61, 4–38-2, 4–34 | 4–28, 4–30-1, 4–30-2, 4–30-4 | 4–31, 4–39 | CAGSTGCAGCTGCAGGAG |
| IGHV5a | | 5–10-1, 5–51 | | | GARGTGCAGCTGGTGCAG |
| IGHV6a | 6–01 | | | | CAGGTACAGCTGCAGCAG |

**Table 2 IGKV and IGLV domain developability and primer sequences.**

| ID | Desirable | Common in natural | Neutral | Poorly developable | Primer sequence |
|---|---|---|---|---|---|
| IGKV1a | 1–39, 1–27, 1–16, 1–12, 1–5 | 1D-39, 1–33, 1D-33 | 1D-17, 1D-16, 1D-12 | 1–17 | RACATCCAGATGACCCAG |
| IGKV1b | 1–09 | | 1–13, 1D-13 | | GMCATCCAGTTGACCCAG |
| IGKV1c | 1–05 | 1–08 | 1D-43 | | GCCATCCRGATGACCCAG |
| IGKV1d | | 1D-08 | | | GTCATCTGGATGACCCAG |
| IGKV3a | 3–20, 3–11, 3–15 | | 3D-20, 3D-11, 3D15 | | GAAATTGTGTTGACRCAG |
| IGLV1a | 1–47 | 1–36, 1–44 | | | GAAATAGTGATGACGCAG |
| IGLV1b | 1–40, 1–51 | | | | CAGTCTGTGCTGACTCAG |
| IGLV2 | 2–11, 2–14, 2–23 | 2–08, 2–18 | | | CAGTCTGTGYTGACGCAG |
| IGLV3a | 3–21 | | | | CAGTCTGCCCTGACTCAG |

To facilitate the assembly of single-chain variable fragments (scFv) we followed our previously established strategy[3]. Briefly, an additional set of primers was used to reamplify the 12 IGHV and the 6 IGKV and 4 IGLV to insert a region of overlap to generate the scFv linker—containing the lox site—and restriction sites to facilitate cloning (PCR2 in Fig. 1c). 24 Individual scFv gene pools were generated by individually assembling each of the 12 IGHV amplicons with a pool of either the 6 IGKV or the 4 IGLV (PCR3 in Fig. 1c). Each of the 24 assembled scFv gene pools was individually ligated into the pDAN5 phagemid vector[3] and separately transformed into electrocompetent TG1 *E. coli* cells to generate what we refer to as the "primary library" with a total number of transformants of ~$4.9 \times 10^9$.

The final recombinant libraries were prepared by recombining the different variable domains as previously described[3] (Fig. 1c). Equal amounts of phage from the 24 primary libraries were combined into two libraries, IGHV-IGKV and IGHV-IGLV. These primary libraries were used to infect Cre-recombinase-expressing bacteria at a multiplicity of infection (MOI) of 200:1. Due to the presence of two non-compatible lox sites in the linker between the variable domains and in the phagemid vector downstream of the g3, the scFv clones present in each bacterium after infection shuffle the IGHV-g3 segment increasing the diversity produced by each bacterium by the square of the number of clones present. This process resulted in the formation of the recombined library, termed the "secondary library", in which phenotype (scFv) and genotype (scFv gene) are not coupled. To re-establish phenotype-genotype linkage, phages produced from the secondary library were then used to infect XL1Blue (non-Cre-recombinase bacteria) at an MOI < 1 to produce the tertiary phage library, which was used for all selections.

**NGS library analysis: general considerations**. Because of the previously observed discrepancy between transformant count and actual diversity[23,25,27], NGS analysis was performed to obtain a more accurate representation of library diversity and V gene-family distribution. We used the common clonotyping strategy of the heavy-chain complementary determining region 3 (HCDR3), considered the most important component in conferring binding activity and specificity, as well as a more comprehensive merged CDR1 + 3 approaches to assess diversity.

NGS has revolutionized the sequencing and analysis of antibody repertoires, revealing that immune repertoires adhere to heavily-tailed power distributions[43], in which there are many low-abundance and a few highly-abundant clones, further complicating complete read coverage and analysis[44,45]. MiSeq has proved to be one of the more popular NGS platforms, able to provide a sequencing depth of up to $2.5 \times 10^7$ paired-end reads of 300 bp, sufficient to cover the entire IGHV, IGKV, or IGLV regions. However, this is still insufficient to cover the entire diversity of massive libraries, and as a consequence novel informatic approaches have been applied to precisely capture sequence diversity in massively undersampled sequence populations[46–48]. The best approach to overcome the undersampling problem is to generate more reads, for which we employed Illumina's NovaSeq sequencing platform. With >100-fold deeper HCDR3 coverage over the MiSeq platform (~$10^7$), NovaSeq (~$10^9$) has the potential to provide an accurate measure, rather than an estimation, of the total diversity even though it comes with other

limitations (read lengths of 150 bp severely limit V gene coverage whereby one loses CDR2 information).

**Sequence analysis approach and clonotyping strategy.** Inherently, errors can be introduced throughout the library construction process, particularly during cDNA synthesis and PCR amplification of V genes. Despite deviating from the donor genetic material, these errors contribute to the diversity of a constructed library if they introduce functional amino acid changes. While unique molecular identifiers (UMIs) or molecular amplification fingerprinting (MAF)[49–54] can be incorporated to reduce biological and technological sampling errors[44,53], these approaches require consensus-building (~3–5 reads per identifier) and thus sequencing depth that greatly exceeds actual library size, making these methodologies most powerful for smaller libraries generated from selections (e.g., <$10^6$ diversity). To overcome these hurdles, we employ a hierarchical clonotyping strategy for singletons similar to that previously described[55–60] using both abundance filtering ($N \geq 2$) and Hamming distances.

Under this scheme, sequences with few read counts (singletons) are considered only if they are more than a given Hamming distance away from sequences with greater abundance ($N \geq 2$). This approach retains singletons that would be lost through abundance filtering alone and would-be essential contributors in a massively undersampled population. After filtering for Phred score (lower quartile, $Q \geq 37$)[61] and optimal V(D)J alignment $E$-value scores ($e$-value $\leq 1.0 \times 10^{-12}$), to eliminate low-quality reads, we obtained a total of $1.82 \times 10^9$, $1.65 \times 10^8$, and $1.13 \times 10^8$ sequence reads for the IGHV, IGKV, and IGLV libraries, respectively, by NovaSeq analysis (Table 3).

**Library diversity.** Final library diversity is reported in Table 3 using increasing Hamming distances or abundance criteria. Measured heavy-chain CDR3 (HCDR3) diversity ranged from $2.2 \times 10^7$–$5.17 \times 10^7$ at a Hamming distance of ≤0–3. We also estimated HCDR3 diversity using a negative exponential accumulation model [91]. Estimated HCDR3 diversity ranged from $2.14 \times 10^7$ ($N \geq 2$) to $7.14 \times 10^7$ (Hamming = 0) (Table 3). The non-saturating species accumulation curve (Fig. 2a) suggests that we are undersampling the population even when approaching $2.0 \times 10^9$ reads. Because functional diversity of an IGHV chain is not restricted to HCDR3 alone, we merged HCDR1 and HCDR3 amino acid sequences to determine diversity at Hamming distances from 0–3. Under this approach diversity of the heavy-chain falls within the range of $4.5 \times 10^7$–$1.5 \times 10^8$ unique sequences (Table 3). Light chain CDR3 (LCDR3) diversity was 10 to 100-fold lower than the HCDR3 with estimated ranges of $1.83 \times 10^5$–$3.04 \times 10^5$ and $6.11 \times 10^5$–$1.10 \times 10^6$, for IGKV and IGLV, respectively. The light chain exhibited a merged LCDR1 + 3 diversity between $7.28 \times 10^5$–$1.95 \times 10^6$ and $6.84 \times 10^5$–$1.96 \times 10^6$ for the IGKV and IGLV libraries, respectively. In contrast to the species accumulation curve of the HCDR3, LCDR3 diversity (IGKV and IGLV combined) saturates at ~$2.0 \times 10^8$ reads converging at ~$7.0 \times 10^5$ sequences at Hamming 1–3 and ~$1.5 \times 10^6$ for a Hamming of 0 (Fig. 2b). Merged LCDR1 and LCDR3 show a measured diversity of just under $7.0 \times 10^5$–$2.0 \times 10^6$ (Table 3). Full-length chain (IGHV or IGKV/ IGLV) ORFs exhibited percent productivity of 89% and 84% for the IGHV or IGKV/IGLV chains, respectively (Supplementary Table 1).

**Primer promiscuity.** While the intended goal was primer-specific amplification of target scaffolds, due to the similarity of V genes

**Table 3 Diversity of the variable regions measured at varying hamming distances.**

| Hamming | # Reads | Unique CDR1 measure[a] | Unique CDR2 measure[a] | Unique CDR3 measure | Unique CDR3 estimate | CDR1 and 3 total |
|---|---|---|---|---|---|---|
| IGHV | | | | | | |
| 0 | 1.82E + 09 | 1.04E + 05 | 2.05E + 05 | 5.17E + 07 | 7.14E + 07 | 1.50E + 08 |
| 1 | 1.82E + 09 | 5.52E + 04 | 1.06E + 04 | 3.24E + 07 | 4.54E + 07 | 1.03E + 08 |
| 2 | 1.82E + 09 | 4.74E + 04 | 8.61E + 04 | 2.67E + 07 | 3.49E + 07 | 8.24E + 07 |
| 3 | 1.82E + 09 | 4.59E + 04 | 8.29E + 04 | 2.20E + 07 | 2.72E + 07 | 7.02E + 07 |
| $N \geq 2$ | 1.82E + 09 | 4.54E + 04 | 8.22E + 04 | 1.49E + 07 | 2.14E + 07 | 4.46E + 07 |
| IGKV | | | | | | |
| 0 | 1.65E + 08 | 3.24E + 04 | 1.41E + 04 | 4.68E + 05 | 3.04E + 05 | 1.95E + 06 |
| 1 | 1.65E + 08 | 1.87E + 04 | 7.78E + 03 | 2.51E + 05 | 1.96E + 05 | 1.01E + 06 |
| 2 | 1.65E + 08 | 1.55E + 04 | 7.00E + 03 | 2.29E + 05 | 1.85E + 05 | 7.89E + 05 |
| 3 | 1.65E + 08 | 1.49E + 04 | 6.90E + 03 | 2.23E + 05 | 1.83E + 05 | 7.45E + 05 |
| $N \geq 2$ | 1.65E + 08 | 1.45E + 04 | 6.85E + 03 | 2.20E + 05 | 2.37E + 05 | 7.28E + 05 |
| IGLV | | | | | | |
| 0 | 1.13E + 08 | 2.20E + 04 | 1.23E + 04 | 1.12E + 06 | 1.10E + 06 | 1.96E + 06 |
| 1 | 1.13E + 08 | 1.21E + 04 | 6.70E + 03 | 6.96E + 05 | 6.67E + 05 | 1.02E + 06 |
| 2 | 1.13E + 08 | 9.69E + 03 | 6.04E + 03 | 6.40E + 05 | 6.20E + 05 | 7.10E + 05 |
| 3 | 1.13E + 08 | 9.15E + 03 | 5.89E + 03 | 6.29E + 05 | 6.11E + 05 | 6.91E + 05 |
| $N \geq 2$ | 1.13E + 08 | 8.92E + 03 | 5.83E + 03 | 6.23E + 05 | 6.57E + 05 | 6.84E + 05 |
| IGHV/IGKV total | | | | | | |
| 0 | – | 3.37E + 09 | 2.89E + 09 | 2.42E + 13 | 2.17E + 13 | 2.93E + 14 |
| 1 | – | 1.03E + 09 | 8.25E + 07 | 8.13E + 12 | 8.90E + 12 | 1.04E + 14 |
| 2 | – | 7.35E + 08 | 6.03E + 08 | 6.11E + 12 | 6.46E + 12 | 6.50E + 13 |
| 3 | – | 6.84E + 08 | 5.72E + 08 | 4.91E + 12 | 4.98E + 12 | 5.23E + 13 |
| $N \geq 2$ | – | 6.58E + 08 | 5.63E + 08 | 3.28E + 12 | 5.07E + 12 | 3.25E + 13 |
| IGHV/IGLV total | | | | | | |
| 0 | – | 2.29E + 09 | 2.52E + 09 | 5.79E + 13 | 7.85E + 13 | 2.94E + 14 |
| 1 | – | 6.68E + 08 | 7.10E + 07 | 2.26E + 13 | 3.03E + 13 | 1.05E + 14 |
| 2 | – | 4.59E + 08 | 5.20E + 08 | 1.71E + 13 | 2.16E + 13 | 5.85E + 13 |
| 3 | – | 4.20E + 08 | 4.88E + 08 | 1.38E + 13 | 1.66E + 13 | 4.85E + 13 |
| $N \geq 2$ | – | 4.05E + 08 | 4.79E + 08 | 9.28E + 12 | 1.41E + 13 | 3.05E + 13 |

[a]CDR1 and CDR2 diversity is estimated from MiSeq with 1.18E + 07, 4.60E + 05, and 2.82E + 05 reads for heavy, kappa, and lambda.

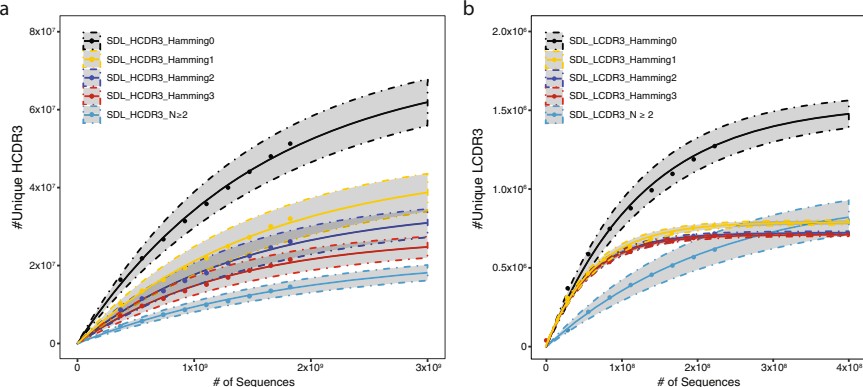

**Fig. 2 Heavy and light chain diversity accumulation curves. a** Species accumulation curves for HCDR3 at varying hamming distance criteria reveal continued accumulation at all edit string distance cutoffs (Hamming 0–3) and using accumulation cutoffs ($N \geq 2$). **b** Species accumulation curves of light chain CDR3 at varying hamming distance criteria reveal saturation of diversity around 7–8 × $10^5$ for the combined heavy and light chain for hamming distances 1–3 and for abundance cutoffs of $N \geq 2$.

and their corresponding primers, we also wanted to assess how well the primers amplified the V families and/or genes of interest. Of the 12 separate IGHV primers used (Supplementary Table 2), the mean V family on-target efficiency was 72.5%. Primer families IGHV5a and IGHV6a showed the greatest promiscuity with average off-target frequencies of 50–69%. Due to the high sequence similarity between genes within the same family (e.g., IGHV1a-d), V-gene amplification exhibited greater promiscuity with a mean on-target amplification frequency of 44.9%. Using nucleotide Hamming distance, we clustered the 12 primer families using Ward's minimum variance unsupervised agglomerative clustering approach.

We defined three predominant gene-family clusters that support the observed on-target/off-target amplification profiles (Fig. 3a). Cross-amplification of IGHV1, IGHV3, and IGHV5 families (Fig. 3b, d, f) can be explained by similar homologies observed among these different primers (cluster 1 in Fig. 3a). The IGHV2 family primers exhibit little similarity to other families and showed the least amount of off-target gene-family amplification (Fig. 3c, cluster 2 in Fig. 3a). IGHV4 and IGHV6a clustered into a single subset (cluster 3 in Fig. 3a) explaining the off-target amplification of IGHV4 amplicons by the IGHV6a primer family (Fig. 3g). The high on-target efficiency of the IGHV4 primer family and minimal amplification of IGHV6a (Fig. 3e) is explained by the alternate strategy employed for the IGHV4 family subset. The promiscuous amplification of IGHV domains not targeted by specific primers (for instance, IGHV5 sequences when IGHV1 was the targeted domain (Fig. 3b)), is due to the fact that some IGHV domains differ by only 0–2 nucleotides in the region targeted by the primers used for amplification (and particularly at the 3′ end). This is particularly important when such "contaminant" domains are fairly abundant in the natural repertoire.

**Clonal dominance.** To confirm that no single HCDR3 population dominates the data set, we looked at the relative abundance of unique HCDR3s clonotyped using our hierarchical strategy detailed above with a Hamming distance restriction 1 across all primer sets (Fig. 4a). All but the IGHV4 and IGHV6 primer set exhibited a clonal dominance of ≤1.0%. The biased clonal dominance of IGHV4 is thought to be a consequence of the alternative strategy used for this particular family, which optimized on-target amplification at the cost of diversity. The low abundance of IGHV6 in natural repertoires likely explains the relatively high relative abundance (~1%) for the most abundant IGHV6 HCDR3 clonotype. Across the primer sets, IGHV1a, IGHV3a, and IGHV5a show

the least clonal dominance, as represented by the higher percentage representation of lower abundance clones. The majority of sequences are represented by singletons, ranging from 58 to 72% for individual primer sets, with 90% of the population represented by HCDR3s present less than three times. Clonal HCDR3 overlap was <2% for all the individual library scaffolds. The most abundant HCDR3 in the combined IGHV population was found to be from the IGHV4 subset, representing ~1.1% of the population, while the remaining clones all showed a frequency of <0.1% (Fig. 4b). Ranking clones by abundance and tabulating the cumulative sum shows that 50% of the population is represented by ~$10^5$ clones, with the remaining 50% represented by >3.2 × $10^7$ sequences for the heavy chain. Ranking copy number of HCDR3 clonotypes by percent representation reveals 68% of the combined population was represented by singletons with 90% of the population represented by an abundance count of ≤16 (Fig. 4c). As expected, the IGKV and IGLV light chains, show greater LCDR3 clonotype dominance, represented 8–9% and 3–4% of the total IGLV and IGKV populations, respectively (Fig. 4d). Fifty percent of the sequence reads are represented by 100 unique sequences, with the remaining 50% represented by 2.5 × $10^5$ and 6.9 × $10^5$ sequences for IGKV and IGLV light chains, respectively. In contrast to IGHV, IGKV and IGLV singleton representations contribute to a minority of the population (~20%), indicative that ≥80% of the population is constituted by 2 or more read counts for both IGKV and IGLV, respectively (Fig. 4e) and reflecting the oversampling of diversity.

**V gene distribution.** IGHV genes identified for their improved developability or increased representation in the naive repertoire are classified in Table 1. Any IGHV genes not highly represented in the naive repertoire, or found in the poor/desirable developability groups, were classified as neutral. As the PCR products generated by each primer set were combined at equimolar ratios ahead of sequencing, we normalized the output to represent equal read count by primer pair (Fig. 4f). The overall scaffold representation is composed of improved or neutral developability or represented by elevated abundance in the naive immune system with few poorly developable scaffolds. Only IGHV1 showed a significant (though still relatively low) representation of poorly developable (~11%) scaffolds due to the presence of the unfavorable IGHV1–2 scaffold. An examination of the normalized population shows that ~44% of the library has scaffolds with highly favorable mAb developability properties. A similar analysis applied to the light chain reveals a predominant representation of highly developable and/or neutral scaffolds. There is a strong representation of developable light chain V genes (IGKV1–9,

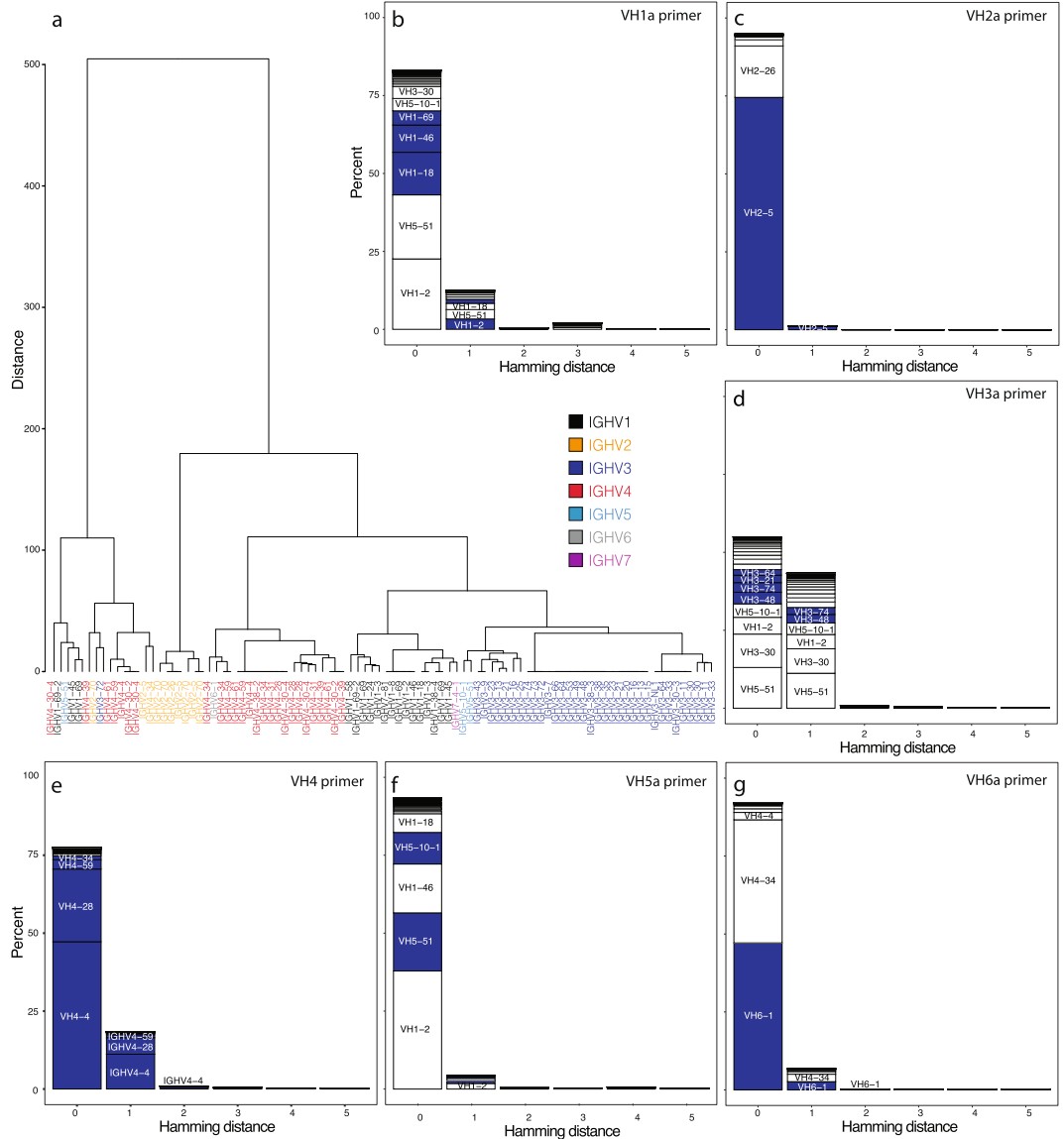

**Fig. 3 Relative cluster of primer sets corresponding to germline sequences for insight into off-target. a** Unsupervised hierarchical clustering of 18 bp region at 5′ portion of the V-region shows three predominant clusters that include many cross-family members. **b** Analysis of the off-target gene-family amplification for IGHV1a as an example of the primer specificity for the IGHV1 family. **c** Analysis of the off-target gene-family amplification for IGHV2a as an example of the primer specificity for the IGHV2 family. **d** Analysis of the off-target gene-family amplification of IGHV3a to represent the primer specificity for the IGHV3 family. **e** Analysis of the off-target amplification for IGHV4. **f** Analysis of the off-target gene-family amplification of IGHV5a to represent the primer specificity for IGHV5. **g** Analysis of the off-target amplification for IGHV6a.

IGKV1–12, IGKV1–38, IGKV1–39, IGKV1–27, IGKV1–5, IGKV3–15, IGKV3–11, and IGKV3–20) at approximately ~68% of the $V_\kappa$ family distribution (Fig. 4g), with a minimal representation of poorly developable scaffolds (<1%). Approximately 23% of the IGLV library is composed of the poorly developable IGLV3-1 population, though the predominating fraction (>45%) contain scaffolds considered to be more optimally developable (Fig. 4h).

**Biophysical diversity of naive library**. The intended goal of the library construction was not only to capture a broad range of unique HCDR3 sequences distributed across developable scaffolds but also to capture a broad range of biophysical properties. With certain V–J regions favoring select binding, our circos graph analysis reveals that the SDL contains a well-mixed population of all possible V–J combinations in both heavy (Fig. 5a) and light

chains (Fig. 5b). Matrix pairing of V/J pairs reveals biased over-representation of IGHJ4, IGHJ5, and IGHJ6, correlating with the intended IGHV frequency distribution IGHJ1/2 (5%) IGHJ3 (10%), IGHJ4/5 (60%), and IGHJ6 (25%) described above.

The SDL captures the full range of HCDR3 and LCDR3 lengths (3–42 amino acids) (Fig. 5c). The median HCDR3 length is 14 ($\sigma = 3.73$), with a lower and upper 10% quantile of 10 and 20, respectively. As expected, a larger proportion of extremely long HCDR3s ≥ 30 is seen within the IGHJ6 antibodies. Light chain CDR3s range from 3–21 to 3–23 for IGKV and IGLV, respectively. Median lengths for IGKV and IGLV are 9 ($\sigma = 1.23$) and 11 ($\sigma = 1.45$) with lower/upper quantile values of 8/10 and 9/12, respectively, reflecting previous trends of IGKV supporting longer CDR lengths. In contrast to IGHV/IGJHJ pairings, IGKV/IGKJ pairings do not show an increasing length trend across IGKJ families or IGKV families, though abnormal

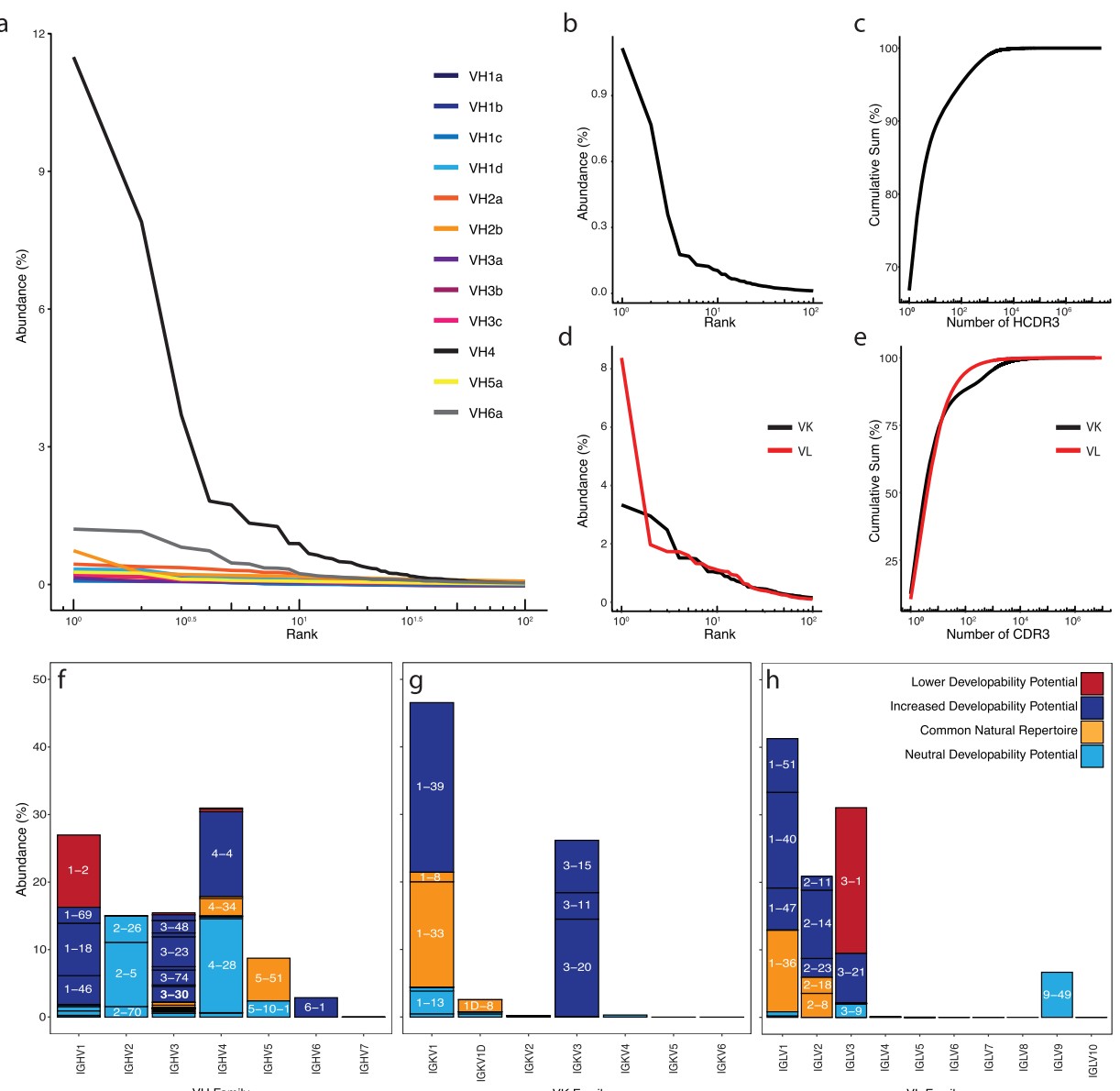

**Fig. 4 Primer pair clonal dominance, overlap and relative diversity, and V-region distribution of heavy and light chain in the final library. a** Cross-family comparison of clonotype dominance by individual primer sets. **b** Relative overlap of the HCDR3 among IGHV1–5 primer sets. **c** HCDR3 clonal dominance, **d** Cumulative clonal rank of HCDR3 assessing % representation by different copies of individual clones (e.g., 1 singletons, 2 duplicates). **e** LCDR3 cumulative clonal rank of the (IGKV or IGLV) LCDR3. **f** LCDR3 clonal dominance of the IGKV or IGLV library. Scaffold representation and Tiller's developability criteria analysis for the **f** IGHV domains, **g** IGKV domains, and **h** IGLV domains.

lengths (≥20) are apparent in IGKV1/IGKJ1, IGKV1/IGKJ3, IGKV1/IGKJ4, and IGKV3/IGKJ2.

We observed a broad and even HCDR3 net charge distribution above and below neutral charge using Lehninger's pK scale at pH 7.0. Median HCDR3 net charge is ~0.00 (±1.39) similar to IGKV-CDR3 at ~0.00 ($\sigma = 1.31$), and slightly greater than IGLV-HCDR3 at −0.82(±1.02). The overall net charge ranges from −6.0/6.0, −3.0/5.0, and −6.0/3.5 for IGHV, IGKV, and IGLV, respectively.

We used the Gravy index score to understand the extent of hydrophobicity and hydrophilicity of given H/LCDR3 amino acid sequence. Sequences with index below 0 display more hydrophilic tendencies, while those above 0 exhibits more hydrophobic profiles. Across the entire spectrum of HCDR3 there is an apparent bimodal distribution with larger peak focalized to hydrophilic indices between 0 and −1.5 (Fig. 5d). A second, smaller peak is noted

across IGHV families within the 0–1 range. Overall median hydrophobicity values for HCDR3 sits at −0.56 ($\sigma = 0.68$) with upper/lower 10% quantile bounds of −1.37 and −0.56, respectively. This bimodal trend is not apparent in light chain CDR3 with $V_{\kappa}$ Gravy index values are distributed around the median value of −1.31 ($\sigma = 0.69$) with upper/lower 10% quantile bounds of −2.11 and −0.43, respectively. Interestingly, $V_{\lambda}$ families show index trends localized around the median of −0.18 ($\sigma = 0.59$) with upper/lower 10% quantile bounds of −0.92 and −0.18, respectively.

**Functionality against therapeutic targets.** We chose four therapeutically relevant antigens to validate the SDL: (1) a tumor necrosis factor (OX40); (2) a T-cell inhibitor (B7-H4); (3) T-cell activation marker (CD40L); and (4) granulocyte-macrophage colony-stimulating factor (GM-CSF). The library was panned

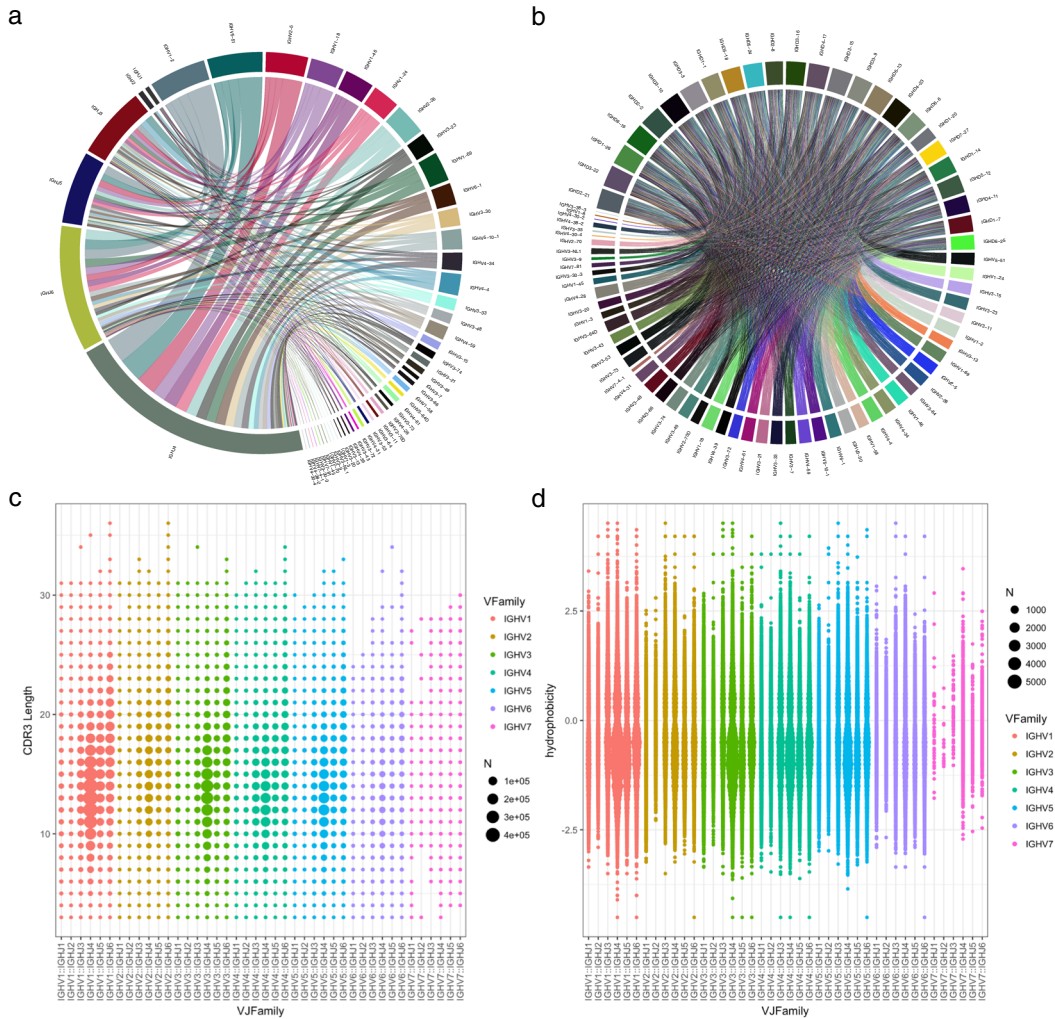

**Fig. 5 Biophysical characteristics in the final library show a broad distribution of binding features. a** Circos graph of V/J coupling in the heavy chain.
**b** Circos graph of V/J coupling within combined IGKV/IGLV library. **c** HCDR3 length distribution for different IGHV:IGHJ combinations. **d** Calculation of
hydrophobicity index using GRAVY for different IGHV:IGHJ combinations.

against the biotinylated antigens bound to magnetic streptavidin
beads at saturation. After two rounds of phage selections, the
resulting populations enriched for binding were transferred to a
yeast-display system and two rounds of sorting were performed
by FACS, as previously described[62]. The final polyclonal popu-
lations displayed on yeast were tested for binding specificity
against all four targets and no cross-reactivity toward the non-
specific targets was observed for any of the selected populations
(Fig. 6a). Additionally, we tested each individual polyclonal
population with decreasing amounts of the antigens to estimate
the detection limit for the selected antibodies (Fig. 6b). For all
four antigens, a binding population was observed when staining
the polyclonal population at 2 nM of antigen. Particularly the
binding activity against GM-CSF was well above background at
2 nM antigen concentrations, indicative of binders within the low
nanomolar to high picomolar affinity range.

Sequence sampling with NGS (MiSeq, ~58,000 reads per
target) revealed a significant number of "HCDR3 clonotypes"
(HCDR3 sequence clusters if Hamming ≤1 and abundance >1)
and "HCDR3-Germline clonotypes" (HCDR3 sequence clusters
if Hamming ≤1, a different IGHV germline is called, and
abundance >1). On average 352 HCDR3, clonotypes and 852
HCDR3-germline clonotypes were identified per target, which
differed by antigen (Supplementary Table 3).

We looked at the sequence-based physicochemical properties
of the selected, unique, populations to elucidate whether the
HCDRs of the libraries differed on an antigen-by-antigen basis.
First, we evaluated the pairwise Levenshtein distances among the
antibody sequences obtained after each selection campaign using
merged HCDRs (Fig. 6c) and HCDR3 alone (Fig. 6d). We found
that each campaign selected antibodies with a broad range of
distinct HCDRs and HCDR3s. The OX40 campaign, while still
selecting a panel of highly diverse antibodies, did have a large
subgroup with a high % identity across the CDR regions.

Next, we observed a bias in the net charge of merged
HCDR1–3 (Fig. 6e) and HCDR3 alone (Fig. 6f) across the
selected antibodies toward their respective targets. In particular,
scFvs selected against OX40 and B7-H4 exhibit a clear population
bias with the former preferring negatively charged CDRs, and the
latter shifted toward neutral to positively charged CDRs.

An assessment of antibody HCDR properties against the
different antigens revealed that the majority of clonotyped
antibodies exhibited favorable hydrophilic properties (Fig. 6g,
h). Again, differences were observed between different selected
populations with antibodies selected against B7-H4 containing
populations that were slightly less hydrophilic while GM-CSF
selected antibody populations showing a larger proportion of
antibodies with more hydrophilic properties.

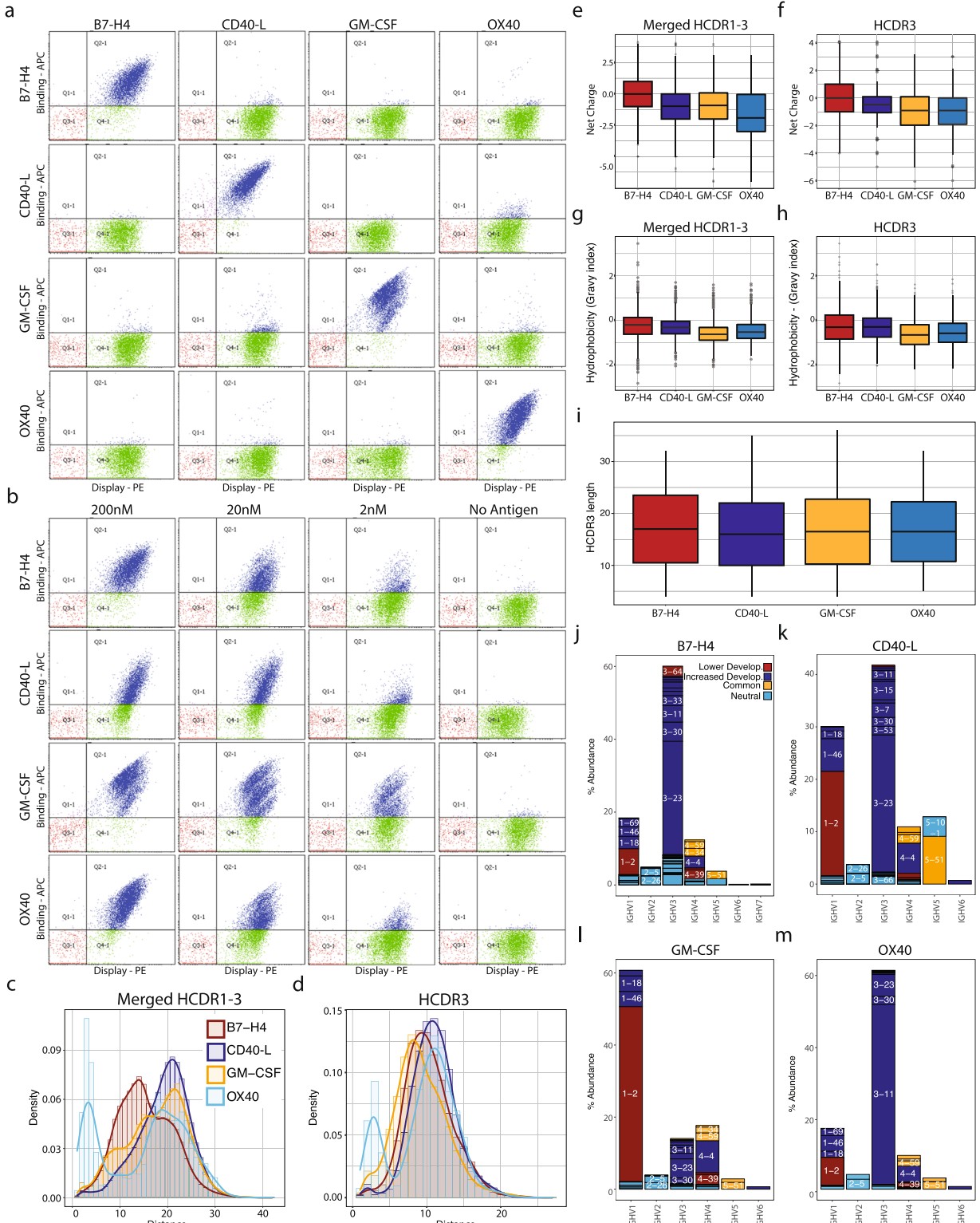

**Fig. 6 Phage and yeast selection against four therapeutically relevant targets shows highly developable candidates and biased biophysical profiles.**
**a** Limited cross-reactivity of yeast scFv across the different panel of antigens at 200 nM. **b** Yeast scFv-display selection against a wide range 2–200 nM of antigen, shows minimal background (0 nM) with a noticeable population obtained from each selection campaign. **c**, **d** Pairwise Levenshtein distances distribution of the merged HCDR1–3 and HCDR3 amino acids. **e**, **f** HCDR3 and merged HCDR1–3 amino acid net charge tabulated using pH = 7 on Lehninger pK scale, showing the clear disparity of binders. **g**, **h** Gravy hydrophobicity index analysis of merged HCDR1–3 and HCDR3 amino acids. **i** HCDR3 length distribution across different antigens. **j**–**m** Analysis of the selected scaffold and their developability profile.

While scFvs showed a broad HCDR3 length distribution from 9 to 20 (Fig. 6i), we observed no clear HCDR3 length bias among the different antigens.

Similar to the net charge and hydrophobicity index profiles above, we observed a clear bias of preferred scaffolds of antibodies selected against respective antigens (Fig. 6j–m). We find that by biasing the library during the construction process to contain a larger proportion of favorable scaffolds, we find we tended to select antibodies with favorable scaffold properties. Using the Tiller et al.[29] criteria, described above, the majority of antibodies selected against OX40, B7-H4, and CD40L contain scaffolds with favorable biophysical properties.

To further show the SDL could be used to isolate actual drug leads, we performed a selection campaign against the therapeutically relevant membrane protein CD73 to isolate specific antibodies and validate success in conversion to conventional IgG. 77 unique clones were identified as specific binders for the recombinant human protein while 54 were cross-reactive for the human and murine forms of the target. No significant liabilities were noticed at the amino acid sequence level of the selected clones, hence, all clones were reformatted and expressed as full-length human IgG1 molecules on a small scale (4 ml culture). 51 of 77 (~66%) human-specific clones were successfully converted and showed binding activity, with a titer, after purification, ranging from 2.6 to over 600 µg/mL and affinities ranging from 71 nM to 156 pM (Fig. 7a). 42 of 54 (~78%) scFv recognizing both the human and the murine proteins were successfully converted into human IgG1, with expression levels ranging from 2.5 to over 600 µg/mL and affinities ranging from 24 µM to 54 pM (Fig. 7b).

## Discussion

The first natural naive phage antibody library described[1] was generated from two donors and was successfully used to select antibodies against a number of different targets, albeit with rather poor affinities. Since then, the general tendency has been to try and use as many donors as possible to maximize the library diversity[63], based on the idea that the immune repertoire, in terms of germline IG loci, varies across human individuals and ethnicities, and this might influence and vary the quality of antibody responses, for instance during exposure to vaccination and disease. For this reason, there has been a tendency to generate libraries from more and more donors, with the idea that such libraries provide both quantitative (more antibodies) and qualitative (antibodies for a wide range of antigen and/or with potential better affinity) advantages.

Here we show that a library carefully crafted from a single donor provides sufficient diversity to select hundreds of antibodies against different targets, notwithstanding the presence of a higher level of clonal dominance when compared to libraries created from multiple donors[22,27]. The disadvantages of such clonal dominance may be offset by a greater percentage of natural IGHV/IGLV and IGHV/IGKV pairings in single-, as compared to multi-, donor libraries, claimed to be advantageous in immune libraries[64]. However, whether the numbers of natively paired antibodies present in a single donor are significant, or indeed relevant, to large naive antibody libraries, remains unclear.

Antibody recognition of a given epitope is mediated by the three complementarity determining regions (CDRs) of each of the heavy and light chains (6 CDRs total). IGHV diversity is generated by the sequential random assembly of ~50 IGHV, 23 IGHV-D, and 6 IGHV-J genes[65–68], with the IGHV-D forming the central core of the binding site. The use of all three IGHV-D reading frames[69,70], recombination between IGHV genes after rearrangement[71], P-nucleotide[72] or N-nucleotide[73,74]-mediated addition, and exonuclease mediated loss[72,75], of nucleotides between IGHV/IGHV-D and IGHV-D/IFHV-J segments further increase diversity. Similar mechanisms apply to the IGKV and IGLV, although there is no D region, and less modification at the junctions. The combination of these diversification strategies with heavy and light chain assortment in an unbiased manner has been estimated to have the potential to generate over $10^{15}$ uniquely expressed B-cell receptors[76,77], far exceeding the estimated ~$10^{11}$ B cells in any individual[78]. Adequate sampling of B-cell diversity remains a problem due to the power-law distribution of sequenced populations (few high abundant clones and many low abundant clones), whereby some rearrangements occurring ≥1000 times more frequently than others[40,79], dominate sequence analyses. This makes sufficient coverage unattainable, a problem aggravated by the short-read lengths of high capacity next-generation sequencing platforms ($2 \times 150$bp with NovaSeq), which miss critical diversity information at the CDR2 region. Even though these issues make accurately measuring (rather than estimating) the true diversity of an antibody library challenging, the application of common standards to this problem should allow the determination of minimum diversity in a manner that allows libraries to be reliably compared and ensures that a given library retains sufficient diversity to select a wide array of putative leads for therapeutic development.

It is not clear how much of such natural diversity can actually be harvested in antibody libraries. In a recent study[26], which

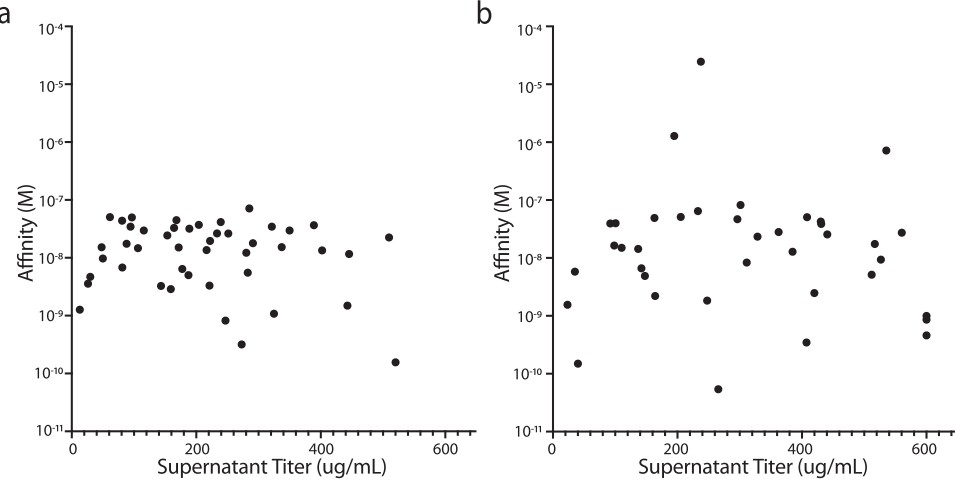

**Fig. 7 Affinities and expression levels of clones selected against human CD73. a** expression level and affinity of human-specific IgG clones **b** expression level and affinity of IgG antibodies selected against CD73 recognizing both the human and the murine proteins.

gathered V genes from 809 different donors, unique reads obtained from 454 pyrosequencing resulted in the identification of only 7373 IGHV and 41,804 IGKV sequences, even though the total number of transformants was ~$10^{10}$ CFU. In another publication[23], also using 454 sequencing, the number of unique IGHV genes was estimated to be ~$2 \times 10^5$, while the number of transformants was $10^5$ fold higher. Finally, a publication[27] combining three different NGS approaches (454, IonTorrent, and MiSeq) determined that a library[3] of $7 \times 10^7$ transformants had an estimated HCDR3 diversity of only ~$3 \times 10^6$. These publications exemplify why colony counting, the usual method used to estimate library diversity, while it may provide a reasonable assessment of combined IGHV/IGLV and IGHV/IGKV diversity, vastly overestimates IGHV diversity. The major problem of using NGS to estimate library diversity is significant undersampling. Even MiSeq (~$10^7$ paired-end reads) does not provide the necessary throughput to generate accurate estimates. In this respect, NovaSeq, as used here, with up to $3 \times 10^9$ paired-end reads may be able to approach the oversampling required for accurate diversity assessments.

Another challenge in estimating naive library diversity is the definition of what constitutes a unique antibody. If antibodies differing by a single amino acid are considered to be different, diversity evaluations will be overestimated due to sequencing errors. Furthermore, functional differences between antibodies differing by single amino acids are usually minimal, unless the amino acid has markedly different biochemical properties and/or located in an area important in binding (e.g., HCDR3). One approach to overcome this problem is the use of clustering, which has been carried out in the past using a variety of methods, including (1) inference through a relationship with unmodified common ancestors[80,81]; (2) hierarchical clustering linkage trees with a variety of mismatches[81]; (3) homology threshold clustering[82–86]; or (4) abundance filtering[87–90]. While it has been suggested that abundance filtering is optimal in many circumstances, Turchaninova and colleagues[49] have pointed out that this type of filtering may be potentially deleterious, particularly in the analysis of naive B-cell repertoires (and for naive libraries as well): when individual clones are present at extremely low frequencies, and diversity exceeds the number of sequencing reads, abundance-based filtering will shrink the observed diversity of naive B cells by eliminating authentic clones sequenced only once. Nonetheless, due to inherent errors introduced through cDNA synthesis, PCR amplification, and Illumina sequencing, a robust clonotyping strategy remains a critical requirement when assessing CDR diversity.

To assess the diversity of the single donor library created here, we use abundance filtering coupled to amino acid Hamming distance analysis. We address Turchaninova's singleton conundrum by including singletons within the analysis: however, "true singletons" are verified only if they fall outside the restrictive Hamming criteria relative to sequences that have undergone abundance filtering ($N \geq 2$). Our rationale for using amino acid Hamming distance is three-fold: (1) library diversity should be based on understanding clone functionality, not underlying nucleotide differences without binding consequences; (2) errors introduced during library preparation, especially at the cDNA and PCR amplification level are "real" sequences in the sense that introduced errors during library preparation become members of the final library, and (3) since most errors introduced during library Illumina sequencing are substitutions, not indels[91] we reason that Hamming (differences at equal length) would be more appropriate than other methods (e.g., Levenshtein distance). With these factors under consideration, we show relative diversities at Hamming distances of 0–3 (corresponding to 100% match, 90–97%, 80–93%, and 70–90% for Hamming 0, 1, 2, 3,

respectively) for CDR3 lengths of 10–30 amino acids, so providing upper and lower bounds for actual sequence diversity. Further, we also employed an abundance filtering cut-off for all unique CDR3 reads of two or more. Applying these methods to the single donor library described here, yields a measured HCDR3 diversity of $5.2 \times 10^7$ at a Hamming 0, which decreases to $2.2 \times 10^7$ at a Hamming of 3. These values increase to $7.1 \times 10^7$ (Hamming 0) and $2.7 \times 10^7$ (Hamming 3) when diversities are estimated using a non-linear regression model[92]. This leads to the conclusion that a significant proportion (>73%) of the true IGHV diversity of natural naive antibody libraries can actually be measured using NovaSeq, regardless of the Hamming distance used. Unfortunately, as NovaSeq is unable to cover HCDR2, it is not possible to determine IGHV germline gene identities. However, taking HCDR1 into account, increases these diversity figures to $1.5 \times 10^8$ (Hamming 0) and $7.02 \times 10^7$ (Hamming 3), reflecting the fact that identical HCDR3 sequence can be encoded by different IGHV and DH genes[93]. The case for the VL chains is more straightforward. As estimated and measured diversities coincide, saturation is reached, and the light chain diversity is measured at $2.2 \times 10^5$ to $4.7 \times 10^5$ for IGKV and $6.3 \times 10^5$ to $1.1 \times 10^6$ for IGLV for Hamming distances 3 to 0, respectively.

One important advantage of using genes based on natural, rather than synthetic, antibodies is the "quality control" carried out by B cells during their development to ensure that antibodies are well expressed. B-cell viability is maintained by tonic signaling proportional to the level of surface-expressed B-cell receptor (BCR)[94]. If levels of surface BCR are too low, RAG, the recombination activating gene, is activated, causing receptor editing (a second rearrangement of the variable light chain that creates a BCR with a new specificity due to the changed light chain). If this new BCR has improved surface expression, further receptor editing is prevented by RAG downregulation[95,96]. Although it has not been shown for B cells, it is well known that antibody display levels on yeast, another eukaryotic organism, reflect the underlying thermostability and expression levels of the displayed component antibodies[97], and directed evolution to improve antibody display on mammalian cells results in improved "developability"[98] Similar mechanisms are also likely to be in play during B-cell development, with the immune system ensuring that antibodies have the potential to be expressed at high levels in plasma cells if required.

In the library described here, we did not use multiplex PCR amplification strategies, which have been shown to bias PCR products[50], but instead used individual 5′ primers with individual 3′ primers. Interestingly, although many of these primers were designed to be specific for particular V genes[31], MiSeq analysis (which allows V gene assignment) of the PCR products obtained revealed that there was a significant degree of promiscuity in the V genes amplified. While some of this was expected, on the basis of sequence similarity (e.g., IGHV1 and IGHV3, IGHV4 and IGHV6), it is notable that all the primers, with the exception of IGHV2, also amplified other V genes. Two factors appear to drive this promiscuous amplification: similarity to the primer and the underlying abundance of a given gene within the cDNA source undergoing amplification. This is particularly striking with the IGHV6 primer, where approximately 50% of the product is IGHV4, a far more abundant gene family.

The library functionality was also proven with successful selection campaigns where several highly specific leads for each of the targets were identified. Moreover, when a therapeutically relevant protein—CD73—was used as a target during a selection campaign a very high number of the selected antibodies, selected as scFvs, were successfully converted into full-length human IgGs retaining some remarkable affinity values. The importance of being able to convert a high number of selected scFv molecules

into full-length immunoglobulins is essential because of the known therapeutic advantages of the IgG format: binding specificity, clinical safety profile, low intrinsic toxicity and immunogenicity, the inherent ability to recruit immune cell function to the site of antigen expression, and the prolonged serum half-life that reduces dosing interval required to elicit a therapeutic effect.

In conclusion, we show here that it is possible to build a highly functional phage antibody library from a single donor, biased towards natural scaffolds with more favorable developability features. Furthermore, by using NovaSeq it is also possible to accurately measure the diversity of such a library. Future use of this approach, combined with common bioinformatic standards, should allow more accurate assessments of antibody library diversities in a manner that will allow different libraries to be compared.

## Methods

**Bacterial and yeast strains.** TG1 (Lucigen): [F′ traD36 proAB lacIqZ ΔM15] supE thi-1 Δ(lac-proAB) Δ(mcrB-hsdSM)5(rK– mK–).

BS1365: BS591 F′ kan (BS591: recA1 endA1 gyrA96 thi-1 D lacU169 supE44 hsdR17) [lambda imm434 nin5 X1-cre].

XL1Blue (Agilent): recA1 endA1 gyrA96 thi-1 hsdR17 supE44 relA1 lac [F′ proAB lacIq.

ZΔM15 Tn10 (Tetr)].

S. cerevisiae EBY100 (GAL1-AGA1::URA3 ura3–52 trp1 leu2Δhis3200 pep4::HIS2 prb11.6R can1 GAL).

**Library generation.** The SDL antibody phage library was created by assembling repertoires of light and heavy-chain variable regions into the scFv format and cloning into the pDAN5 display vector[3]. Total RNA was prepared from $7 \times 10^8$ CD19$^+$ cells purified from an entire single LeukoPak using the Straight From Leukopak CD19 (Miltenyi), with mRNA purified using the "Oligotex mRNA spin-column" kit (Qiagen). The mRNA was used to generate cDNA with Super Script IV (ThermoFisher) and primers specific to the constant region of IgM heavy chains and κ and λ light chains[31].

Twelve heavy and 10 light chain 5′ primers and 4 heavy and 8 light chain 3′ primers, based on previously published sequences[31], were used to generate 450 separate primary PCRs using the cDNA as template. The V genes were reamplified to add an overlapping region corresponding to the scFv linker as well to facilitate the restriction enzyme digestion for the cloning in the display vector.

To maintain diversity, the 12 individual IGHV primary PCRs were PCR-assembled either with the pooled IGKV or IGLV PCR repertoires (as previously described in ref.[3]), and subsequently cloned in 24 separated sub-libraries for a total number of $\sim 5.0 \times 10^9$ clones. We carried out 24 individual phage productions to generate the "primary" phage libraries. Those were combined in two phage pools, IGHV-IGKV and IGHV-IGKV, by mixing equal amounts of phage, and the two IGHV-IGKV and IGHV-IGKV mixes underwent lox-based-recombination[3] to obtain the final tertiary phage library used in the selections.

**Bioinformatics pipeline.** All paired-end sequence reads derived from MiSeq were received from SeqMatic and merged using Paired-end fastq processing using PandaSeq[99]. Due to limitations on the NovaSeq platform, paired-end assembly is bypassed. All sequences were processed by eliminating low-quality reads quality using the FASTX toolkit from the Hannon lab (http://hannonlab.cshl.edu/fastx_toolkit/). We set a threshold whereby 90% of sequences had a minimum Phred[61,100] quality score of ≥25. All clonotyping is performed on our in-house linux, R, and python-base software running using linear processing on a computing cluster composed of 256 nodes of 4× Quad Core AMD Opteron CPUs with 32GB RAM/node and 1TB disk/node. Sequences are annotated to germline V regions using the IgBlast command-line tool[101], which uses the BLAST scoring algorithm[102] to align query sequence to germline IGHV and IGKV/IGLV sequences from the international ImMunoGeneTics database® (IMGT®) database[103]. Appropriate V gene alignment calls were kept that retained an e-value less than $1 \times 10^{-14}$ and a nucleotide percent identity match across the framework and CDR germline regions of ≥90% (alignments do not include CDR3). After appropriate alignment amino acid sequences were subject to Hamming distance clonotyping strategy at 0–3 edit string distances for sequences of the equal length according to the following equation:

$$d^{\mathrm{HD}}(i,j) = \sum_{k=0}^{n-1} \left[ y_{i,k} \neq y_{j,k} \right]$$

where $d^{\mathrm{HD}}$ is Hamming distance of objects $i$ and $j$ with $k$ index of given variable evaluating $y$ from the total number of variables $n$. All data are presented at Hamming distance of 1, unless otherwise noted.

Unsupervised hierarchical clustering performed using R hclust system, set to find relationships among the n observations, or 18 bp sequence dissimilarity is calculated using Ward's minimum variance cluster agglomeration method. This particular approach minimizes total within-cluster variance, where each iteration step pair clusters with minimum between-cluster distances.

The diversity index was calculated according to the following general equation:

$$^{q}D = \left( \sum_{i=1}^{R} p_i^{q} \right)^{\frac{1}{1-q}}$$

where $p_i$ is the proportion that clone $i$ contributes to total sequence population and $q$ takes on different values such as richness ($i = 0$), Shannon entropy ($i = 1$), or Simpson index ($i = 2$), with all estimates taken at $q = 0$, unless otherwise noted. Estimates for species accumulation curve are based on the non-linear negative exponential function.

Biophysical properties were calculated in R using subset of internal packages. Net charge is based on Lehninger's pK scale at pH 7.0. Hydrophobicity is assessed using GRAVY index using the Kyte and Doolittle scale[104].

**Phage and yeast antibody display selections.** Antibody clones displayed as scFvs were isolated by combining phage- and yeast-display methodologies as previously described[62,105]. Briefly, the SDL phage library ($10^{12}$ cfu) was used in 2 rounds of selection on recombinant human proteins using the automated Kingfisher magnetic bead system (ThermoFisher Scientific), applying $2 \times 10^7$ Streptavidin-conjugated magnetic beads (Dynabeads M-280) coated with the biotinylated proteins (100–400 nM), washed and incubated with the antibody phage library. After removal of non-binding phage, the remaining phage particles were recovered from the beads by acid elution and used to infect F′ pilus-carrying bacteria, and amplified by growth overnight at 30 °C. A second selection cycle was also performed; the first round of selections provided a $5 \times 10^4$ to $6 \times 10^5$ cfu while the outputs of the second round of selections provided $1.2 \times 10^6$–$5 \times 10^6$ cfu. After the selection cycle was repeated, the pool of scFv clones was PCR amplified with specific primers introducing an overlapping region with the yeast-display vector pSpec yeast-display vector (based on pDNL6[62], but with N terminal rather than C terminal display as found in pDNL6). The starting material for the PCR amplification consisted of 1 ng of plasmid DNA obtained from the second round of selections, consisting of more than $10^8$ copies of the template, abundantly covering the diversity of the final round of phage selection.

The cut vector and amplification products were co-transformed into competent yeast cells to allow cloning by homologous recombination. The number of clones in each of the yeast mini-libraries covered at least ten times the diversity of the phage selection outputs. The yeast mini-libraries were further enriched for target-specific binders by applying two rounds of flow cytometry sorting, where at least $5 \times 10^6$ cells were screened (FACSAria, Becton Dickinson), as described[62,105]. Up to 10,000 individual yeast cells with both positive antigen-binding signal and scFv display were sorted and propagated for further rounds of selection (gating strategy explained in Supplementary Fig. 1).

**IgG reformatting, expression, and purification.** The scFv sequences of interest were used to generate IgG1 gene constructs and cloned into the mammalian expression vector pCDNA3.4 (ThermoFisher). Small scale expression (4 ml media) was performed for each clone and the expressed antibodies were protein G purified.

**Reporting summary.** Further information on research design is available in the Nature Research Reporting Summary linked to this article.

## Data availability

The data and data sets generated and analyzed during the current study are available on reasonable request by contacting the corresponding author Andrew Bradbury (abradbury@specifica.bio).

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

## Acknowledgements

We thank Dr. Camila Leal-Lopes for her critical reading and advice.

## Author contributions

Study design and supervision: S.D., M.F.E., A.R.M.B., and H.G.N. Writing: F.F., M.F.E., S. D., and A.R.M.B. Antibody library generation: S.D. Sequencing, bioinformatic, data handling, and analysis: M.F.E. and A.A.T. Antibody selections: F.F., L.N., and R.B. Antibody validation: S.M.S.

## Competing interests

M. Frank Erasmus, Sara D'Angelo, Fortunato Ferrara, Leslie Naranjo, and Andrew R.M. Bradbury are employees and stockholders of Specifica Inc. Rebecca Buonpane, Shaun M. Stewart, and Horacio G. Nastri are employees and stockholders of Incyte Corp. The remaining authors declare no competing interests.
