## [Peer Review File · Communications Biology]

Reviewers' comments:

Reviewer #1 (Remarks to the Author):

Bradbury and colleagues have conducted a comprehensive and thorough analysis of sequence diversity metrics from an in vitro antibody library constructed from a single donor. They implemented a targeted PCR approach focused on amplifying favorable V segments, with higher developability profiles, and use high-throughput NGS characterize the library. To add functional relevance to their work, they show that the library can be used to identify diverse sets of antibodies against several different target antigens. This manuscript is clearly written from beginning to end. I endorse this manuscript for publication in Communications Biology. From my perspective, only minor edits are needed.

General:

1. The authors' data support the claim that a library made from a single donor captures significant diversity. While I don't disagree with the conclusion, I am curious whether they believe they would observe important quantitative and qualitative differences in libraries constructed from two different individuals. For example, it is known that not all individuals utilize particular gene segments at the same frequency within their naïve repertoire; this would include genes with desirable developability properties, as well as common in natural and neutral Abs. If particular IGHVs are needed in order to characterize an "ideal" in vitro library, they may not be present in every sample at appreciable levels. Even the breadth of IGHV representation in this library seems somewhat limited. Perhaps this is worth mentioning in discussion section (?).

Minor:

1. Line 34: typo, "validated"
2. Line 34: typo, "therapeutic relevant"
3. Line 58: typo, "candidates"
4. Line 161: I would not use the term "families" here. IGHV4-4 and IGHV4-59 are "genes" within the IGHV4 subfamily (as are IGHV4-31/39).
5. Table 1 (and elsewhere): Gene names "1-f", "4-b", "3-d", etc. are no longer used. These genes have been mapped and assigned new permanent names (IGHV1-69-2, IGHV4-38-2, IGHV3-38-3)

Reviewer #2 (Remarks to the Author):

Bradbury and Coworkers used a single donor to create an antibody library that can be used for screening against various antigens. They showed that for obtaining an antibody repertoire with a very high diversity, thus enhancing the likeliness of obtaining high affinity binders, it is not necessary to combine blood samples from numerous donors. Rather, the way how the library is generated is of major importance aimed at avoiding any bias in gene amplification and clone generation. The major value of that manuscript lies in the fact that the authors describe in detail, how the library was made. They validated its quality by successful screening against several targets by combining state of the art phage and yeast display technologies. This manuscript certainly is of value for the many laboratories in academia and pharmaceutical industry working on antibody isolation. Hence I recommend publication. At some point the paper would benefit, if more experimental details would be provided:

The authors state: After two rounds of phage selections, the

resulting populations enriched for binding were transferred to a yeast display system and two rounds of sorting were performed by FACS, as previously described. How many eluted phagemid genomes were used for yeast display library generation? How many yeast clones were generated? Did the authors intend and achieve oversampling upon gene transfer to yeast? How many clones were eventually screened? What was the size of the yeast mini libraries?

Li 696: mammalian expression vectors. Which ones?

Table 6: What is the difference between line 1 and 2?

Figure 7 A and B: axis labelling missing.

Figure 8: are favorable scaffolds better expressed and reveal higher affinity? This could be indicated by dot coloring.

Typos:

Line 230 Table.1) should read Table 1

Levenshstein should read: Levenshtein

"Twelve heavy and 10 light chain 5' primers and 4 heavy and 8 3' primers, based on". Add "light chain" after 8.

"across the different panel of antigen at 200nM antigen" should read antigens.

Reviewer #3 (Remarks to the Author):

Thank you for inviting me and providing the opportunity to review the manuscript submitted by: M. Frank Erasmus et al. entitles: "A single donor is sufficient to produce a highly functional in vitro antibody library".

The manuscript addresses an important question in the field of Human Antibody Library Construction and highlights the importance of the Antibody Library Size/diversity in isolating therapeutically important leads for human applications. The use of a single donor repertoire to build a diverse and functional library distinguishes this research work from similar prior publications such as Glanville et al 2009 and Kim et al 2017. What is quite intriguing in this manuscript is incorporating criteria important for improving the developability of isolated antibodies including aggregation, solubility, and protease degradation as previously reported and emphasized by Tiller and colleagues in 2013.

The research work is novel and follows a rational experimental design followed by in-depth analyses of the obtained data and at the end testing the value of the constructed library from a single healthy donor to isolate antibodies with the best possible frameworks for therapeutic applications. The rational of this work, the data presented, and the discussion part are scientifically sound and will be appreciated by readers.

Although authors have nicely shown that a single donor scFv library is sufficient in isolating antibodies

with valuable properties but in order to draw a general conclusion on the subject, it would have been best if a multi-donor library could also be built under the same conditions and then the diversity and developability could be compared. Therefore, drawing a final conclusion needs to take into account this fact. I also found that the HAL library work by Kugler et al 2015 is missing in the reference list which highlights the importance of immunoglobulin gene usage among different individuals, in particular, with ethnic/geographical backgrounds. This would be important if the end user of the library are interested to isolate robust anti-pathogen binders which may require a certain gene-family usage. A second point to be considered is in regard to the size of the library and I was looking to see if the authors have elaborated what would be the approximate size of the tertiary library and if the approach used to make this library did not affect the final diversity? Other than these points, I believe the manuscript deserve publishing with minor modifications as highlighted above.

January 12th 2021

Point-by-Point Reply to the reviewers' comments

Reply to Reviewer #1

We are very pleased that Reviewer #1 found our manuscript clear, significant and that he favorably suggested for its publication.

Here are our answers to the reviewer's comments:

1. The authors' data support the claim that a library made from a single donor captures significant diversity. While I don't disagree with the conclusion, I am curious whether they believe they would observe important quantitative and qualitative differences in libraries constructed from two different individuals. For example, it is known that not all individuals utilize particular gene segments at the same frequency within their naïve repertoire; this would include genes with desirable developability properties, as well as common in natural and neutral Abs. If particular IGHVs are needed in order to characterize an "ideal" in vitro library, they may not be present in every sample at appreciable levels. Even the breadth of IGHV representation in this library seems somewhat limited. Perhaps this is worth mentioning in discussion section (?).

As reviewer 1 mentioned, the goal of the present manuscript consisted in demonstrating that the careful design and construction of an in vitro antibody library can effectively replicate and reproduce the diversity of the antibody repertoire of a single donor from which valuable recombinant antibodies can be isolated. We agree with the reviewer that the immune repertoire, including the use of germline IG loci, varies across human individuals and ethnicities, and this may vary the quality of antibody responses during exposure to vaccination and disease, and consequently the quality of in vitro libraries derived from them. For this reason, antibody libraries are usually generated from multiple donors, with the idea that the greater potential diversity is able to generate more antibodies against more targets with potentially better affinity. However, as there is significant redundancy in the immune system, we wanted to explore whether it would be possible to produce a functional library from a single donor, the goal of this study. The strategy presented in the present manuscript was also used in the construction of a multi donor antibody library derived from 10 individual donors, and we hope to report the data obtained by comparing the two libraries shortly in a further publication. We have further commented on this aspect in the discussion section.

1. Line 34: typo, "validated"
2. Line 34: typo, "therapeutic relevant"
3. Line 58: typo, "candidates"

Thank for to the reviewer for noticing those typos that have been addressed in the manuscript.

4. Line 161: I would not use the term "families" here. IGHV4-4 and IGHV4-59 are "genes" within the IGHV4 subfamily (as are IGHV4-31/39).

The reviewer is correct, and we eliminated the term families and replaced it with genes.

5. Table 1 (and elsewhere): Gene names “1-f”, “4-b”, “3-d”, etc. are no longer used. These genes have been mapped and assigned new permanent names (IGHV1-69-2, IGHV4-38-2, IGHV3-38-3)

Thank you to the reviewer for noticing the error. We have fixed the text.

Reply to Reviewer #2

We extremely appreciate that the reviewer recognize that our work is of value for laboratories in the academia and in the pharmaceutical industry.

Here are our answers to the reviewer’s comments:

1. The authors state: After two rounds of phage selections, the resulting populations enriched for binding were transferred to a yeast display system and two rounds of sorting were performed by FACS, as previously described. How many eluted phagemid genomes were used for yeast display library generation? How many yeast clones were generated? Did the authors intend and achieve oversampling upon gene transfer to yeast? How many clones were eventually screened? What was the size of the yeast mini libraries?

In the material and method section we added more details regarding the outputs obtained after the phage selection steps and the sub-cloning of the phage selection outputs into yeast. Moreover, we added more details regarding the number of clones screened and the size of the yeast mini-libraries.

Li 696: mammalian expression vectors. Which ones?

The vector used was pCDNA3.4 from ThermoFisher.

Table 6: What is the difference between line 1 and 2?

We are glad the reviewer noticed the missing information in the Table, we also notice that we forgot to add the legend of Table 6 in the text. The text has been fixed: line 1 shows the number of “Unique HCDR3 clonotypes” (HCDR3 sequence clusters if Hamming ≤ 1 and abundance > 1) while line 2 shows “Unique HCDR3-Germline clonotypes” (HCDR3 sequence clusters if Hamming ≤ 1 , a different IGHV germline is called, and abundance > 1).

Figure 7 A and B: axis labelling missing.

We added axis labelling to the two panels of figure 7.

Figure 8: are favorable scaffolds better expressed and reveal higher affinity? This could be indicated by dot coloring.

The aminoacidic sequences of the selected clones did not show any significant liabilities, so we were not able to really identify better scaffolds in this particular set. We added this observation into the text.

Typos:

Line 230 Table.1) should read Table 1

We fixed the typo

Levenshstein should read: Levenshtein

We corrected the typo.

“Twelve heavy and 10 light chain 5’ primers and 4 heavy and 8 3’ primers, based on”. Add “light chain” after 8.

We followed the reviewer’s suggestion

“across the different panel of antigen at 200nM antigen” should read antigens.

We followed the reviewer’s suggestion

Reply to Reviewer #3

We are glad that the reviewer found our work interesting and scientifically sound.

Here are our answers to the interesting points raised by the reviewer:

1. Although authors have nicely shown that a single donor scFv library is sufficient in isolating antibodies with valuable properties but in order to draw a general conclusion on the subject, it would have been best if a multi-donor library could also be built under the same conditions and then the diversity and developability could be compared. Therefore, drawing a final conclusion needs to take into account this fact. I also found that the HAL library work by Kugler et al 2015 is missing in the reference list which highlights the importance of immunoglobulin gene usage among different individuals, in particular, with ethnic/geographical backgrounds. This would be important if the end user of the library are interested to isolate robust anti-pathogen binders which may require a certain gene-family usage.

Many libraries have been created and published using multiple donors. The scope of the present manuscript was not to suggest that a single donor library can provides a “universal” antibody discovery tool (although it may), or to compare a multi- to a single-donor library, but to emphasize how a solid cloning strategy is able to harvest the full potential of the humoral immune system and isolate valid antibody leads against a set of targets. The present work is focused on those key aspects, while a future manuscript where libraries generated using the same strategy but derived either from a single donor or a set of multiple donors are compared is planned.

We thank the reviewer for suggesting the reference, which has been added to our manuscript. We also added a further statement where we emphasize that the strategy described should be used to generate a multi donor library consisting of donors from different ethnic and geographical backgrounds. Moreover, the present strategy can be used to generate single patient immune libraries, that should be suited to select specific antibodies against a disease or pathogen.

A second point to be considered is in regard to the size of the library and I was looking to see if the

authors have elaborated what would be the approximate size of the tertiary library and if the approach used to make this library did not affect the final diversity?

We have only been able to make approximations of tertiary library size, since the increased diversity arises from shuffling of different VH and VL regions that may not have come together during cloning. In the primary library we obtained 4.9×10^9 transformants, corresponding to 4.5×10^7 to 15×10^7 unique VH (HCDR1+HCDR3), 7.3×10^5 to 19.5×10^5 unique IGKV and 6.8×10^5 to 19.6×10^5 unique IGLV library sequences. Consequently, the maximum primary library diversity is the number of transformants (4.9×10^9), while the theoretical tertiary library diversity is the product of the VH and VK/VL diversities (3×10^{13} to 3×10^{14} for each of IGKV and IGLV), which is in turn limited by the number of bacteria infected when phenotype and genotype are coupled. Given that the volume used was 1 Liter, and the number of bacteria at an OD of 0.5 is 5×10^{11} per Liter, the maximum tertiary diversity is 5×10^{11}). As seen from these calculations, the use of a single donor does not limit the final diversity, since the theoretical diversity easily exceeds the attainable practical diversity.

REVIEWERS' COMMENTS:

Reviewer #1 (Remarks to the Author):

I am satisfied with the authors' responses.

Reviewer #2 (Remarks to the Author):

The authors addressed all critical points that were raised by the the reviewers appropriately. The manuscript can now be recommended for publication

Reviewer #3 (Remarks to the Author):

Thank you for revising the manuscript and incorporating the comments from all reviewers which deemed to be either necessary or important to improve the quality of the manuscript.